# Unsupervised Plaque Segmentation on Whole Slide Images

**Johann Engster**[1,2]    JOHANN.CHRISTOPHER.ENGSTER@IMTE.FRAUNHOFER.DE

**Nele Blum**[2]    BLUM@IMT.UNI-LUEBECK.DE

**Tobias Reinberger**[3,4,5]    TOBIAS.REINBERGER@UNI-LUEBECK.DE

**Pascal Stagge**[1,2]    PASCAL.STAGGE@IMTE.FRAUNHOFER.DE

**Thorsten M. Buzug**[1,2]    THORSTEN.BUZUG@IMTE.FRAUNHOFER.DE

**Jeanette Erdmann**[3,4,5]    JEANETTE.ERDMANN@UNI-LUEBECK.DE

**Zouhair Aherrahrou**[3,4,5]    ZOUHAIR.AHERRAHROU@UNI-LUEBECK.DE

**Maik Stille**[1]    MAIK.STILLE@IMTE.FRAUNHOFER.DE

[1] *Fraunhofer IMTE, Research Institution for Individualized and Cell-Based Medical Engineering*

[2] *University of Lübeck, Institute of Medical Engineering*

[3] *University of Lübeck, Institute for Cardiogenetics*

[4] *DZHK (German Centre for Cardiovascular Research), Partner Site Hamburg/Kiel/Lübeck*

[5] *University Heart Centre Lübeck*

**Editors:** Under Review for MIDL 2023

## Abstract

Atherosclerosis is a multifactorial disease in which deposits of fat form in the arteries. These plaques can cause ischemic heart disease or other follow-up diseases. To investigate etiology and possible treatment options, mice were used as models and histological whole slide images (WSI) stained with Oil-Red-O (ORO) were obtained and analyzed. Currently, the plaque content is often estimated using a threshold-based segmentation technique, which requires a manual selection of reference points. To improve this process, an unsupervised segmentation technique is developed using the W-Net architecture. The network weights are updated using two loss functions, the soft N-cut loss, and a reconstruction loss. The update procedure of both U-networks and the weighting function in soft N-cut loss are adapted to the given task. Since no ground truth is available, the results were compared with a post-processed threshold segmentation. The evaluation showed that a linear decaying pixel distance weighting achieves the highest score. The results indicate that an unsupervised learning procedure is able to correctly identify the plaque clusters.

**Keywords:** Atherosclerosis, Unsupervised Segmentation, Plaque Segmentation, WSI

## 1. Introduction

Atherosclerosis is the main underlying cause of cardiovascular disease (CVD), which is the leading cause of death in industrial nations. So-called plaques form in the arteries, which can lead to CVD-related diseases like ischemic heart disease, thrombosis, or stroke. To further study the initiation and development of the disease as well as fundamental medication and treatment options, mouse studies are being conducted at the Institute for Cardiogenetics, University of Lübeck. A simple threshold-based method is currently used to identify plaque clusters. However, this is limited in its applicability and depends on manually selected parameter settings. Unsupervised segmentation for WSIs has been investigated multiple times (Faust et al., 2020; Fouad et al., 2017), as human bias can be reduced in the training data. An unsupervised segmentation approach proposed by Xia and Kulis in 2017 uses an architecture called W-Net. Both, encoder and decoder, consist of a full U-Net (Ronneberger et al., 2015). Thus, the latent space has image dimension and provides the output segmentation mask, while the decoder output of the W-Net reconstructs the input image. The W-Net training procedure was adapted and applied to the WSIs for plaque segmentation.

## 2. Methods

For the training process, two loss functions are used to update the network weights. Both, encoder and decoder, are updated using the original $J_{\text{reconstruction}}$ loss proposed by Xia and Kulis. However, the proposed $J_{\text{soft N-cut}}$ encoder loss

$$J_{\text{soft N-cut}}(V, K) = K - \sum_{k=1}^{K} \frac{\sum_{u \in V} p\left(u = A_k\right) \sum_{u \in V} w(u, v) p\left(v = A_k\right)}{\sum_{u \in V} p\left(u = A_k\right) \sum_{t \in V} w(u, t)},$$

where $w$ measures the similarity between two pixels, K is the number of classes, and $p\left(u = A_k\right)$ or $p\left(v = A_k\right)$ measure the probability of $u$ or $v$ belonging to $A_k$, was adapted to the given task. Since plaques may exist nearby, the exponential term is replaced by a linear decaying term. in this way, the distance penalization of the default $w$ is reduced. The new $w$ is given by

$$w_{ij} = \exp(-\|F(i) - F(j)\|_2^2/\sigma_I^2) * \begin{cases} 1 & \text{if } d = 0 \\ 1/(\|X(i) - X(j)\|_2^2) & \text{else if } d < r \\ 0 & \text{else} \end{cases},$$

where $X(i)$ is a spatial location, $F(i)$ is the pixel value of node $i$, and d is the pixel distance $\|X(i) - X(j)\|_2$. In addition, $\sigma_I$ and $\sigma_X$ are hyperparameters controlling the degree of penalization. The radius $r$ defines the range of the weighting. A second weighting, called intensity weighting, ignores the pixel distance entirely. Finally, a third weighting combines the default weighting with the intensity weighting

$$w_{ij} = \exp(-\|F(i) - F(j)\|_2^2/\sigma_I^2) * \begin{cases} \exp(-\|X(i) - X(j)\|_2^2/\sigma_X^2) & \text{if } d < r_1 \\ 1 & \text{else if } r_1 < d < r_2 \\ 0 & \text{else} \end{cases},$$

using two radii $r_1$ and $r_2$. The networks can be updated in different ways by the loss functions. For instance, the authors of the original publication suggest a sequential update of encoder and decoder. However, a combined update of both networks proved to be more stable and was therefore used for further experiments.

## 3. Experimental Results

The networks were trained on data provided by the Institute for Cardiogenetics consisting of 1103 ORO-stained WSIs from 104 different mice. Data was split 60/40 for training and testing. The number of classes is set to 5 to reduce bias, as a minimum of 3 classes (white background, artery, and plaques) are expected inside the WSIs.

Table 1 shows the mean IOU for the different weightings compared to the threshold-based plaque segmentation for classes $c = 0, 1, 2$, sorted from highest to lowest. The classes $c = 3, 4$ were near-empty for all approaches and are therefore ignored. For all tested methods, multiple classes achieve a plaque mean IOU $> 0$. If for example, the class corresponds to the artery, it can achieve a mean IOU $> 0$ as the plaques are inside it.

Table 1: Mean IOU for classes $c = 0, 1, 2$, compared to the threshold-based plaque mask.

| Weighting | $c = 0$ | $c = 1$ | $c = 2$ | Mean |
|---|---|---|---|---|
| default | 0.26 | 0.15 | 0.00 | 0.14 |
| linear | **0.41** | 0.12 | 0.01 | **0.18** |
| intensity | 0.24 | 0.19 | 0.00 | 0.14 |
| two radii | 0.24 | **0.22** | **0.05** | 0.17 |

Figure 1 shows exemplary linear weighting masks for the classes $c = 0, 1, 2$. The red-appearing plaques are correctly segmented, with some still remaining artifacts.

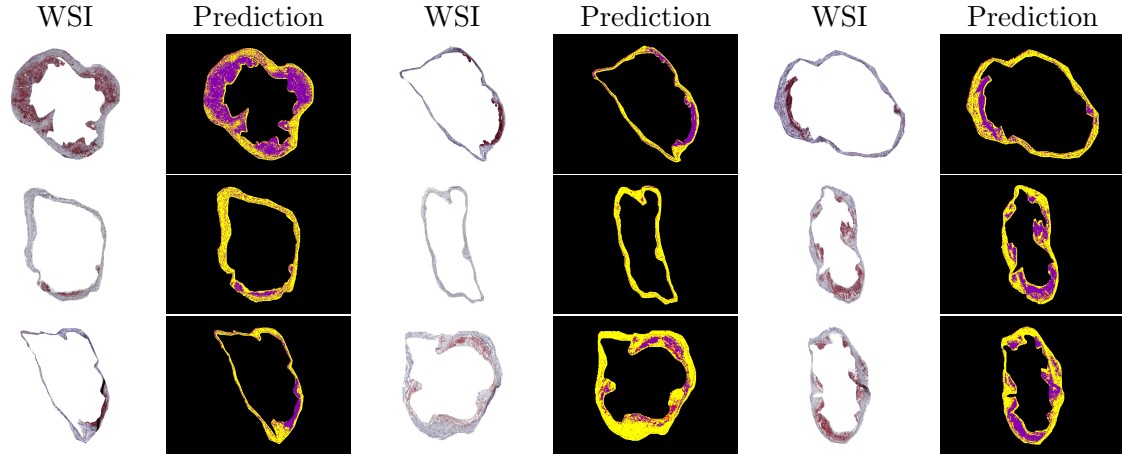

| WSI | Prediction | WSI | Prediction | WSI | Prediction |

Figure 1: Qualitative linear weighting W-Net examples. Class 0 = ■, 1 = ■, and 2 = ■.

## 4. Conclusion

The results show that the W-Net is able to detect the plaques as a stand-alone class. The tested linear and two radii weightings achieve higher scores than the default weighting. The intensity weighting achieves the lowest consensus, while the linear decaying weighting achieves the highest. In the future, suitable post-processing, a detailed statistical validation, and analysis of the different weightings and the update order are required.

## Acknowledgments

The authors would like to thank Annett Liebers, Maren Behrensen, Petra Bruse for technical support. We are also grateful to the members of the Erdmann laboratories for feedback and discussions. This work was supported by the German Federal Ministry of Education and Research (BMBF) in the context of the German Centre for Cardiovascular Research (FKZ 81Z0700108, FKZ 81X2700133).

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
