# OpenReview forum: "Unsupervised Plaque Segmentation on Whole Slide Images"
_MIDL.io/2023/Short_Paper_Track — MIDL 2023 Short paper track Poster_

### Official Review · Reviewer_dzun · 2023-04-22
**Limited novelty and insufficient experiments**

**Rating:** 5
**Confidence:** 4

**Review:**

Summary:

This paper explores the use of unsupervised segmentation to identify plaque clusters in whole slide images (WSIs) for investigating atherosclerosis. The proposed method adopts the W-Net architecture which utilizes two loss functions: a soft N-cut loss and a reconstruction loss. The results are compared to a post-processed threshold segmentation shows that the proposed method can correctly identify plaque clusters.

Weakness:

1. The novelty of this paper is limited. The proposed method is mainly based on previous work [1] and only different weighting strategies are tested.
2. The experiments are insufficient. The proposed method should be compared to other baseline unsupervised segmentation methods to show its effectiveness.


[1] X. Xia and B. Kulis. W-Net: A Deep Model for Fully Unsupervised Image Segmentation, 2017.

Strength:

1. The qualitative results are good, which shows that W-Net is able to detect the plaques with high precision.

---

### Official Review · Reviewer_4ck3 · 2023-04-24
**Unsupervised Plaque Segmentation on Whole Slide Images**

**Rating:** 6
**Confidence:** 4

**Review:**

This paper presents an approach based on unsupervised segmentation to identify atherosclerotic plaques in histology of coronary arteries from a large cohort of mice.
The slides are stained with a special staining that highlights the plaque in red, compared to the rest of artery tissue.
The unsupervised approach is based on an existing method, and authors propose modifications to some terms to account for application-specific constraints.

PROS
Unsupervised learning approaches are interesting because they overcome the limitation of time consuming manual annotations.
This work presents (to the best of my knowledge) a fairly novel application to segmentation of atherosclerotic plaques in whole-slide images.
Proposed modifications to the loss function seem to be somehow effective when considering some classes.
The number of cases used is interesting, with >1,000 whole-slide images used.

CONS
It looks like the authors consider a special staining (its name is not mentioned though), which stains the plaque in red, which in my opinion simplifies the segmentation problem substantially. In fact, authors achieve decent baseline results with simple thresholding.
Other changes in the configuration of the model lead to some improvement, but no configuration improves segmentation for all classes.
Three classes are mentioned but no clear labels, authors say artery, plaque, white background, but including black there are 4 colours in the segmentation output, where I guess black is white background, then what is the red color?